# Topology, Vegetation and Stratigraphy of Far Eastern Aapa Mires (Khabarovsk Region, Russia)

Stanislav Kutenkov [1], Vladimir Chakov [2] and Viktoriya Kuptsova [2,*]

1   Institute of Biology, Karelian Research Centre of the Russian Academy of Sciences, 11 Pushkinskaya, 185910 Petrozavodsk, Russia; effort@krc.karelia.ru
2   Institute of the Water and Ecology Problems, Khabarovsk Federal Research Center of the Far Eastern Branch of the Russian Academy of Sciences, 56 Dikopoltseva, 680000 Khabarovsk, Russia; Chakov@ivep.as.khb.ru
*   Correspondence: Victoria@ivep.as.khb.ru; Tel.: +7-421-232-57-55

**Abstract:** Aapa mires (string-flark fens) are one of the main types of mires in northern Eurasia. It has an almost continuous distribution from Scandinavia to Kamchatka, disappearing in continental climate areas and becoming one of the dominant types in more oceanic zones. This article first presents the topological features of string-flark aapa, their vegetation and peat stratigraphy related to different elements of microrelief at the southernmost borders of boreal mires of cryolithozone (51–52 N), in the Lower Amur region (Russia). String-flark fens are very similar to the aapa mires originally reported for the European North. The waterlogged minerotrophic central fen, with a ribbed surface pattern, is surrounded by oligotrophic bogs. The mosaic structure of the vegetation cover in the fens is determined by microtopography: mesooligotrophic dwarf shrub–herb–sphagnum strings, mesoeutrophic herb–sphagnum lawns, and sparse herb cover in water flarks. The flora, for the most part, corresponds with the European aapa, and has some characteristics of eastern features. We relate the localized evolution of string-flark complexes with water basin hydrology changes. The formation of string-flark complexes in pre-existing fens was preceded by the cessation of river flood waters over the surface of the mires. The further decline of erosion has led to the development of dwarf shrub–sphagnum communities containing microrelief. The immature strings of the aapa mires attest to the ongoing active change processes of the mires.

**Keywords:** aapa mire; geography of mires; mire vegetation; Lower Amur region; river basins; peat stratigraphy

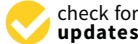



## 1. Introduction

Aapa mires (string-flark fens) are one of the main types of mires in northern Eurasia. They were first identified by Cajander [1], and have a distinctive surface pattern and specific flora composition. Although understanding of aapa mire has widened [2–6], here we follow a narrow understanding of the term aapa which remains in Russia. The main features of aapa mires, according to T. K. Yurkovskaya [7], are as follows: a concave surface; the presence of heterotrophic string-flark complexes in the concave area of mire massif; eu- or mesotrophic peat deposits in the central parts; water-mineral nutrition through atmospheric, diluvial and ground waters; and a combination of herb, sphagnum and hypnum synusia cover. There is also a substantial flow of water due to the strong incline of mires in the area. Aapa is found in both independent massifs and those areas where transition fens among raised bogs are a part of complex mire systems. In the south of the European range of aapa mires remains are only those that a parts of mire systems [8].

Patterned fens have a pan-boreal range [2,7–10], which has been confirmed by analysis of satellite images. In Eurasia, aapa mires occur in forest tundra, and northern and middle taiga, from the west of Scandinavia to the Pacific coast [7,10,11]. Nevertheless, the representation of aapa mires by longitudinal (phytogeographical) sectors of Eurasia and their degree of exploration varies substantially [8].

In the European North, aapa mires are widely spread from the Arctic Ocean coast to latitude 61° N, near the southern boundary of the middle boreal subzone [11,12]. They reach their maximum development in the northern part of Fennoscandia, where aapa is the dominant zonal type of mire massif [2,9,13]. Here, they were first described by Cajander [1], and for Russia by Zinserling [14], and are generally well studied. Aapa mires receive water-mineral nutrition both through underground waters and through snowmelt water from the surroundings [12,14]. Traditionally, three types of aapa mire massifs, differing in morphology and composition of their vegetation, are distinguished within the territory of European Russia [7,9,15]: northern European forest-tundra (Lapland), boreal Fennoscandian (Karelian ring) and north-eastern European (Onega-Pechora).

In west Siberia, the typical aapa mires are found in various natural vegetation zones from the forest-tundra to the south boreal zone [16–18] and, sometimes, sub-boreal forest [19]. According to E.D. Lapshina [18], their development is related first to the peculiarities of water-mineral nutrition and the hydrologic regime and, to a lesser extent, to climate. There are scarce data on the aapa mires of central Siberia, and no data on aapa mires in east Siberia [8].

On the east coast of Eurasia, aapa mires are well-known to be in the Kamchatka region. In Koryakia, the northeast Asian forest tundra, herb-sphagnum-hypnum (aapa) is present [20,21], with dwarf shrub-sedge-sphagnum strings and herb-hypnum flark-pools. There are also aapa mires rich in diversity southward, on the Kamchatka Peninsula. In the north, they are combined with palsa mires [22,23], which are located along the river valleys and the seacoast. In the eastern, ocean zone, they become a dominant type of mire [24,25]. In the western part of the peninsula, which is exposed to the Okhotsk Sea, blanket bogs are dominant, while the herb-hypnum-sphagnum (aapa) mires are present to a smaller extent in the seaside river valleys [25,26]. Aapa mires on the southern tip of Kamchatka Peninsula are common [23]. Formed by regular volcanic ashfall conditions, they are notable for eutrophication, relatively low developed moss cover where oligo- and mesotrophic sphagnums are not present, vascular plants, and shallow deposits with ash layers—they are referred to as special south Kamchatka aapa mires. Information on the location of aapa mires in Khabarovsk Territory and the Amur River region in general has not been reported until recently. Thus, Y.S. Prozorov, the founder of the Russian Far East mire science school, points out that the aapa mires zone here discontinues [27], and the flat and domed palsa mire zone when moving southwest is immediately replaced with a zone of "heterotrophic" mires. The latter are sphagnous heterotrophic mires of mixed and atmospheric nutrition with homogeneous tussock microrelief, widely found in the Middle Amur lowland.

The description of aapa mires given in this article—with their peculiar transverse pattern, flora and stratigraphy of deposits—is the first provided for the Khabarovsk region in particular, and the Amur River region in general.

On the Eurasian continent, the Russian Far East differs from the rest of Siberia and the western regions of Russia. During the Pleistocene age, there was a large glacier in this region on the shelf of the eastern sector of the Arctic with elevation points around 900 m. This fact, as well as the absence of a strait between the Arctic Ocean and the Pacific around the ancient Beringia, substantially hardened the water exchange between the world's oceans and the Arctic, in essence, creating an enclosed body of fresh water [28,29]. In addition, due to the presence of mountain glaciers in Yakutia, Chukotka, Kamchatka in Southern Priokhotye (the Okhotsk Sea region), and the Lower Amur region, for many thousands of years the water course has been collecting alluvial sediments as layered structures of sub-surface rocks and ice blisters. On most of them, after the warming that occurred during the Holocene, peat deposits began to form. These peat deposits, in the territories of the Lower Amur region with southern borderers at the latitude of around 50° N, which have well-known thermal insulation, have until now been preserved permafrost. Through this process, organic-cryogenic forms of meso- and microrelief are frequently present inn pattern bogs—the thalweg parts of the valleys of the ancient water courses buried by peat deposits.

As the permafrost continued to thaw, multi-directional vibrating block movements [30,31], as well as continued active development of floodplain processes caused by sudden abundant summer precipitation, have led to continued erosion, and, as a result, to the development of dynamic mire-forming processes and the active transformation of the surface of mires. String-flark complexes, which develop in fens may also be a consequence of these processes of hydrology change [32–35].

The research objectives, in addition to the publication of the first data on aapa-mires noted for the region, are to compare the vegetation composition with similar mires known from other regions of Eurasia, as well as to identify the dynamics of mire vegetation, ways of formation and directions of further development of string-flark complexes.

## 2. Materials and Methods

### 2.1. The Study Area

The mires studied are located in the Evoron-Chukchagir depression (EChD) in the central part of the Khabarovsk region, stretching from north to south. According to phytogeographical zoning, the area of study belongs to the northern subzone of coniferous forests, the Okhotsk-Kamchatka floristic province (Evoron-Chukchagir plain district) [36,37]. According to the mire zoning diagram by Y.S. Prozorov [31], it is here where the border of two zones lies: heterotrophic sphagnum mires of mixed coniferous-broad-leaved forests and palsa mires of the southern and middle boreal forest zone. The mires occupy a significant area of the depression, remaining almost unstudied until now. Preliminary data on the mire formation process of the area have only been presented by Y.S. Prozorov thus far [30,31]. At the same time, in the territory of the depression, complex mire systems have developed, combining massifs of raised bog, transitional and floodplain riverside mire types. These systems contain smaller lakes and mineral meander cores with forests of larch and birch. Some sites have string-flark complexes directed along run-off streams, similar to those described earlier for the Khabarovsk region [38–40], and string-flark complexes of transverse direction, which are the main focus of this study.

The formation of the relief of the alluvial surface mainly took place during the Pleistocene, against the backdrop of multiphase mountain glaciation on the system of ridges of Yam-Alin, Dusse-Alin and Badzhal. Between the first two and the latter was the valley of the river Amgun, with tributaries and an extensive system of channels. Y.F. Chemekov [41] notes not less than four glaciations on Yam-Alin ridge. The last of them, the one of Seletkan, is clearly observed here by the end of Wurm. Its final stage during the preboreal period was accompanied by two warmings of short duration: the Bolling and the Allerod. The first one is dated during the period of 14,700–14,100 yr BP, a period between the earlier and middle Dryas. The second one is dated during 13,900–12,680 yr BP, which is in the interval between the middle and older Dryas. During the Bolling interstadial, a period of almost 400 years, the air temperature in the northern hemisphere could rise to values consistent with the Boreal or Atlantic periods of the Holocene [42]. For the Lower Amur region and the Southern Priokhotye, this period was marked by an intensive melting of mountain glaciers. This phenomenon resulted in excessive water content of the rivers Tugur and Amgun, including their tributaries. Moreover, in especially wet periods, the overflow was large enough to provoke crossflows to other catchment basins. Thus, on the flattened water divide of Tugur-Nimelen, the water-abundant Tugur not only flowed over to the Nimelen valley, but also formed quite a large water reservoir that existed until the end of the boreal period, when it filled up with sapropel and transformed into a modern raised bog [43]. The consequences of another overflow, the River Amgun, are obvious in the upper parts of the valleys of the rivers Dosmi and Taksakan, and the right tributaries of the Evur river flowing into the Evoron Lake. Today, due to erosion along the Amgun river valley, although it can no longer be seen, it is clearly evident in the swampy valleys of the above rivers by the eutrophic and eu-mesotrophic mires formed in the thalweg declines of the valleys. This fact is proved by the absence of permafrost on such swampy areas, while outside the warming

influence of groundwater, sphagnum oligotrophic bogs are widespread here, peat deposits of which thaw only by 15–20 cm, and no earlier than mid-August.

### 2.2. Climate

The climate in the area is extremely continental. The average annual temperature is −3.2 °C. The non-frost period lasts 100–130 days [44]. The winter is frosty (the average monthly temperature in January is −28.4 °C), the summer is warm and sunny (the average monthly temperature in July is 17.9 °C). The active cumulative temperature (during the period with average daily temperatures above 10 °C) amounts to 1704 °C. Within a year, there is 453 mm of precipitation on average, with 75% falling on liquid fraction [45]. The most precipitation (45%) falls in June–August. Annual evaporation amounts to approximately 400 mm. Although the region is located in an area of excessive moisture (precipitation outpaces evaporation), one of the main sources of peatland moisture is the melting of permafrost. The water content of mires oftentimes reaches their maximum during flood periods, which occur in the summer-autumn period [46] when the mountain snowfields melt and seasonal precipitation occurs.

### 2.3. Study Sites

Studies were conducted on the mire massifs southwest of Lake Chukchagir: eastward of Mount Amukan (A1, 51°46′44.61″ N, 136°10′27.03″ E), eastward of the road to Lake Chukchagir where it crosses a large mire (A2, 51°46′48.66″ N, 136°11′51.92″ E), and on the mire at the stow of Armali (A3, 51°50′24.01″ N, 136°19′31.30″ E) (Figure 1).

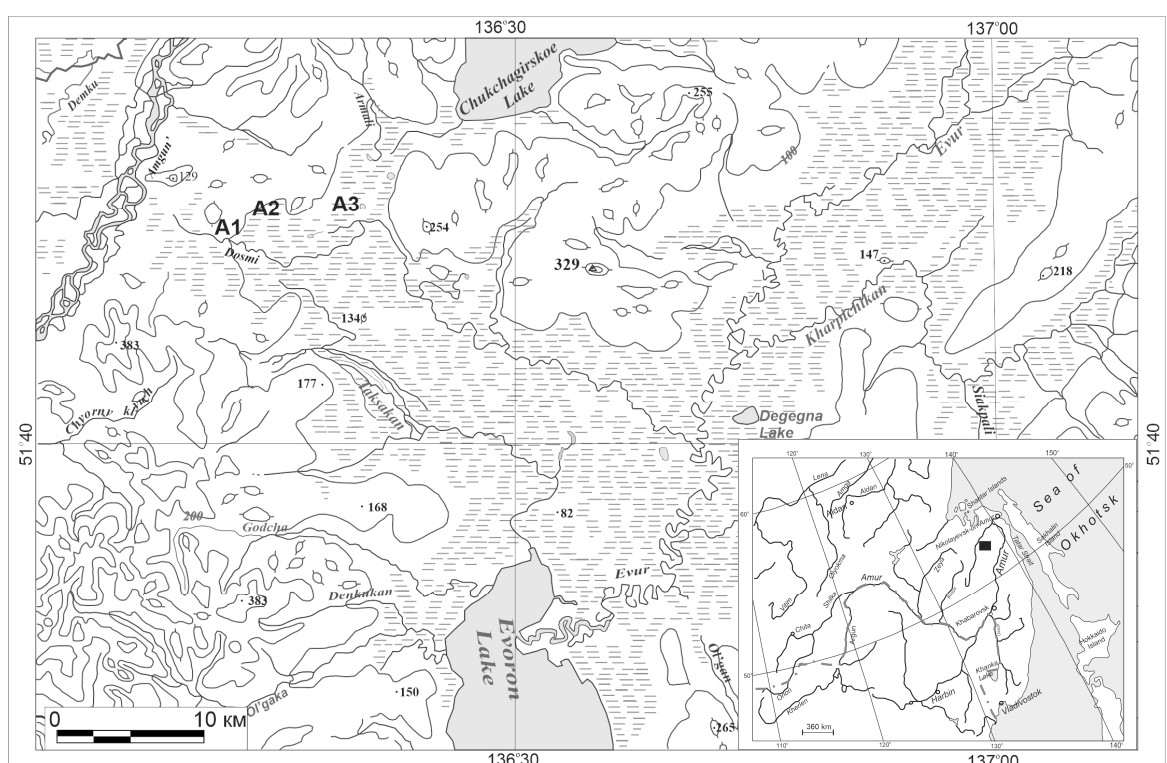

**Figure 1.** Study sites: A1–A3.

The studies were conducted by applying a route method and describing the vegetation on the sample plots (SP). On each massif there were three to six SP, within which geobotanical relevés were made according to a standard technique [47], with determination of species composition and coverage (%) of species for individual layers of vegetation. Owing to the microrelief-associated heterogeneity of the vegetation, relevés were made for the two to three main forms of microtopography (strings, lawns and flarks) on each

SP within their natural contours. The area of relevés sites varied from 20 to 400 m². There were 14 SP determined in total—11 of them were in the central areas of the mires within the string-flark complexes (SFC), and three on the adjacent plots with smoother relief. This study analyzes the SP of SFC only, with 32 separate relevés received in total.

The peat depth of the SP was measured by applying a Russian peat borer ("torfyanoj bur Instorfa"). To reconstruct the genesis and dynamics of the mires and certain microrelief elements, on the plot with the maximum peat depth, a layer-by-layer peat sampling was conducted to determine the botanical macrofossil composition. The peat samples were taken from the string all the way down the deposit and from the flark down to 2 m. Twenty-one peat samples from the string and 10 samples from the flark, at 10–30 cm increments, were taken to observe peat stratification.

## 3. Results

### 3.1. Topology

The fen sites with SFC are located in the central parts of the mires. They have an elongated shape along the massif, and are, as a rule, limited by *Sphagnum*-dominated raised bogs with a more even surface. The border between them is indistinct. The SFC themselves are perpendicular to the general decline of the mire massif, which is clearly shown on high-resolution satellite images (Figure 2). For A1, the mire with SFC occupies a larger area of the massif, having the width of 400–600 m, while the bogs appear as narrow forested margins up to 100 m wide. The SFC of A2 are placed within a complex mire system and are 100–150 m wide. On A3, they are also represented in a fen up to 500 m long, stretching over the bog with a total width of 4 km. The peat deposit of marginal bogs has permafrost at a depth of 50 cm. In some places, it is also observed under the wide strings in the marginal parts of SFC.

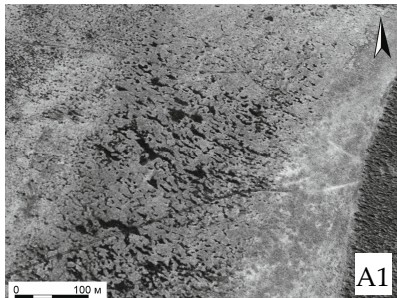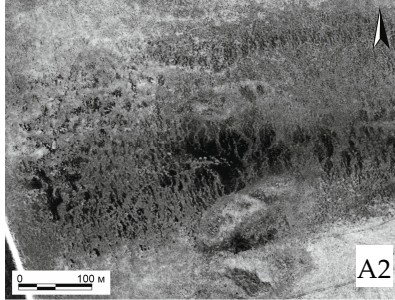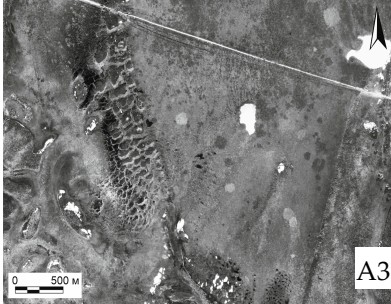

**Figure 2.** Satellite image of string-flark patterns. A1–A3 sites.

There are three basic topological levels presented in the studied complexes: strings 25–50 cm high (on average 35 cm); flarks with water 10–30 cm deep; and the intermediate level presented by low (5–10 cm) cushions, limbs of high strings and low strings and hillocks (hereinafter, the lawns). The ratio between elements, their forms and size vary. The strings are composed of dwarf shrubs entangled stems, on which moss loosely "hang". Their surface can easily be flattened down to the water level. In some cases, the positive landforms (strings) appear as interlinked chainlets of high sedge hillocks. As a rule, they dominate in the microrelief forming a peculiar, netted pattern. In the A3 site, strings have an arched form joining in cascades, and the flarks have a roundish or even rectangular shape. The strings are 3–15 m and, occasionally, 20 m wide. The flarks are 5 to 200 m long, with a width of 2 to 80 m. Here, extensive flarks are sometimes found deeper pools without vegetation (Figure 3).

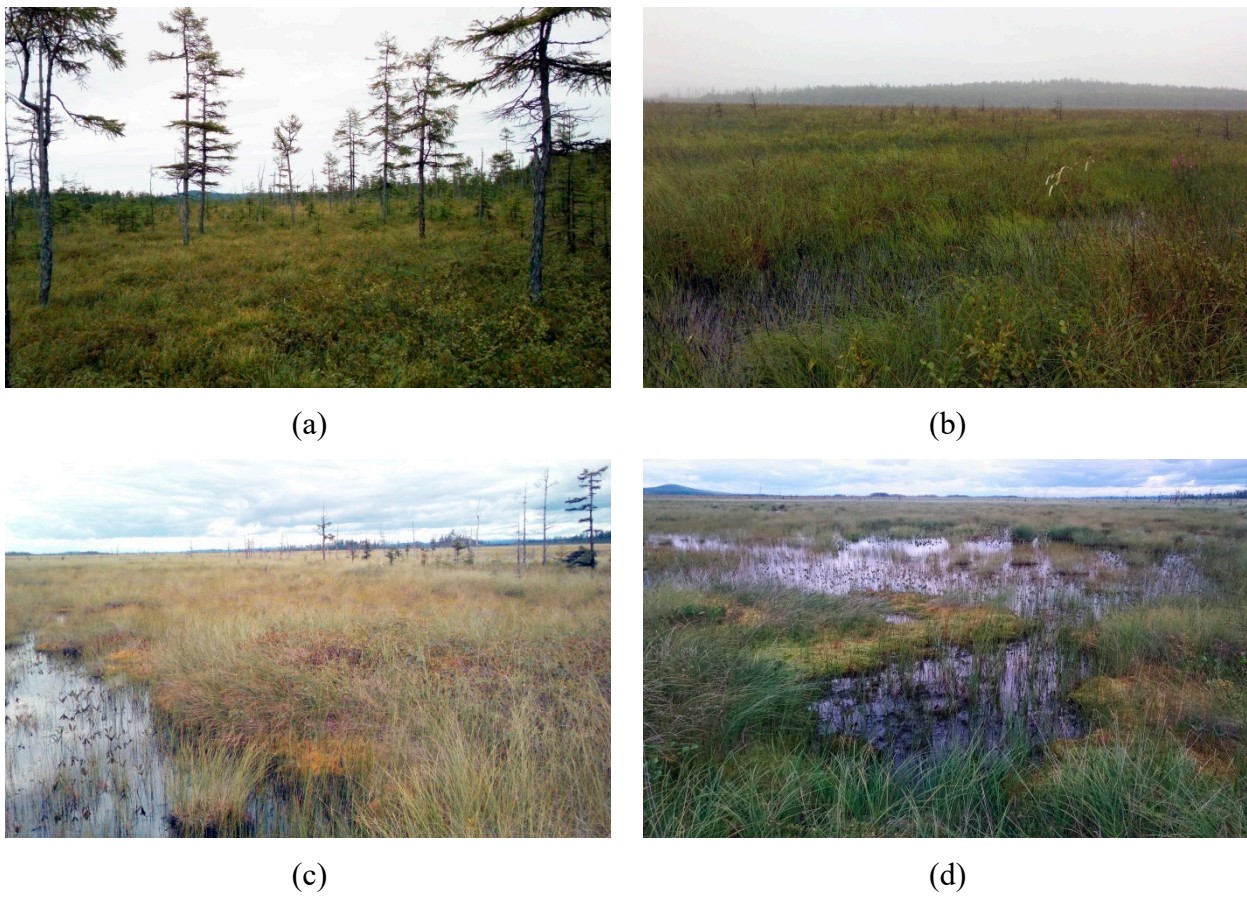

(a)

(b)

(c)

(d)

**Figure 3.** An overall view of the study mires: (**a**) Marginal Larix bog; (**b**) String-flark complex with narrow flarks, A1 site; (**c**) String-flark complex with wide flarks, A3 site; (**d**) Water flark, surrounded by *Sphagnum papillosum* lawns, A3 site.

### 3.2. Vegetation

The vegetative cover of the mires is heterogeneous according to the topology. The high strings are occupied with herb-dwarf shrub-sphagnum communities, the lawns with herb-sphagnum communities, and the flarks with herb ones. The vegetation of the same topological levels between the three mire massifs is relatively similar, which allows for the following data summary (Table 1).

**Table 1.** Phytocoenotic record of aapa complex microforms.

| | Species | String, Height * of 25–50 cm | Low Hillock, Lawn, Height of 5–25 cm | Flark |
|---|---|---|---|---|
| | Number of relevé | 12 | 9 | 11 |
| | **Vascular Plants** | | | |
| 1. | *Larix cajanderi* Mayr | V$^1$ ** | I$^+$ | |
| 2. | *Betula divaricata* Ledeb. | V$^{10}$ | IV$^+$ | I$^1$ |
| 3. | *Andromeda polifolia* L. | V$^5$ | V$^3$ | V$^+$ |
| 4. | *Calamagrostis langsdorfii* (Lik.) Trin. | IV$^+$ | II$^+$ | I$^+$ |
| 5. | *Carex cespitosa* var. *minuta* (Franch.) Kük. | I$^+$ | II$^+$ | IV$^2$ |
| 6. | *Carex chordoriza* L. | II$^+$ | II$^+$ | II$^+$ |
| 7. | *Carex gynocrates* Warnst. | I$^+$ | | |
| 8. | *Carex lasiocarpa* Ehrh. | IV$^5$ | V$^{10}$ | V$^2$ |
| 9. | *Carex laxa* Wahleb. | | | III$^+$ |
| 10. | *Carex limosa* L. | I$^+$ | IV$^+$ | V$^+$ |

**Table 1.** *Cont.*

| | Species | String, Height * of 25–50 cm | Low Hillock, Lawn, Height of 5–25 cm | Flark |
|---|---|---|---|---|
| 11. | *Carex middendorffii* F.Schmidt | III² | I⁺ | I⁺ |
| 12. | *Carex tenuifolia* Wahlehb. | I⁺ | | I⁺ |
| 13. | *Chamaedaphne calyculata* (L.) Moench | V⁵ | III⁺ | |
| 14. | *Drosera anglica* Huds. | | III⁺ | V⁺ |
| 15. | *Drosera rotundifolia* L. | IV⁺ | IV⁺ | |
| 16. | *Eleocharis wichurae* Boech. | | | I⁺ |
| 17. | *Equisetum fluviatile* L. | IV⁺ | IV⁺ | V² |
| 18. | *Eriocaulon schischkinii* Tzvelev | | II⁺ | IV⁺ |
| 19. | *Eriophorum gracile* Koch | I⁺ | | |
| 20. | *Eriophorum russeolum* Fr. | I⁺ | | I⁺ |
| 21. | *Glyceria spiculosa* (F. Schmidt) Roshev. | IV⁺ | III⁺ | |
| 22. | *Habenaria linearifolia* Maxim. | I⁺ | III⁺ | |
| 23. | *Hammarbia paludosa* (L.) O. Kuntze | | II⁺ | |
| 24. | *Iris laevigata* Fishet Mey. | III⁺ | V⁺ | V⁺ |
| 25. | *Juncus stygius* L. | | III⁺ | IV⁺ |
| 26. | *Ledum palustre* L | III¹ | | |
| 27. | *Lobelia sessilifolia* Lamb. | II⁺ | IV⁺ | V⁺ |
| 28. | *Menyanthes trifoliata* L. | IV⁺ | V⁺ | V⁵ |
| 29. | *Osmundastrum asiaticum* (Fernald) Tagawa | I⁺ | | |
| 30. | *Ostericum maximowiczii* (F. Schmidt ex Maxim.) Kitag. | I⁺ | | |
| 31. | *Oxycoccus microcarpus* Turcz. ex Rupr. | III⁺ | I⁺ | |
| 32. | *Oxycoccus palustris* Pers. | V⁴ | IV⁺ | I¹ |
| 33. | *Parnassia palustris* L. | II⁺ | III⁺ | I⁺ |
| 34. | *Pedicularis grandiflora* Fisch. | I⁺ | | |
| 35. | *Pedicularis resupinata* L. | I⁺ | | I⁺ |
| 36. | *Pogonia japonica* Rchb. f. | II⁺ | + | |
| 37. | *Rhynchospora alba* (L.) Vahl | I⁺ | V² | V² |
| 38. | *Salix myrtilloides* L. | III⁺ | III⁺ | |
| 39. | *Sanguisorba parviflora* (Maxim.) Takeda | V¹ | IV¹ | I⁺ |
| 40. | *Saussurea amurensis* Turcz. ex DC. | III⁺ | II⁺ | |
| 41. | *Scheuchzeria palustris* L. | | IV⁺ | V¹ |
| 42. | *Scutellaria regeliana* Nakai. | II⁺ | III⁺ | |
| 43. | *Smilacina trifolia* (L.) Desf. | IV⁺ | II⁺ | |
| 44. | *Sparganium hyperboreum* Laest. ex Beurl. | | | I⁺ |
| 45. | *Spiranthes sinensis* (Pers.) Ames | I⁺ | II⁺ | |
| 46. | *Triadenum japonicum* (Blume) Makino | | | I⁺ |
| 47. | *Trichophorum alpinum* (L.) Pers. | | III⁺ | |
| 48. | *Utricularia intermedia* Hayne | | | V¹¹ |
| 49. | *Utricularia macrozhiza* Le Conte | | | III⁸ |
| 50. | *Utricularia minor* L. | | | III⁺ |
| | **Bryophytes** | | | |
| 51. | *Aulacomnium palustre* (Hedw.) Schwägr. | III⁺ | II⁺ | I⁺ |
| 52. | *Campylium stellatum* (Hedw.) CEOJensen | | II⁺ | II⁺ |
| 53. | *Dicranum bergerii* Bland. Ex Hoppe | I⁺ | | |
| 54. | *Dicranum bonjeanii* De Not. | | I⁺ | |
| 55. | *Dicranum polysetum* Sw. | I⁺ | | |
| 56. | *Drepanocladus polygamus* (Bruch. et al.) Hedenäs | | | I⁺ |
| 57. | *Pleurozium schreberi* (Brid.) Mitt. | II⁺ | | |
| 58. | *Polytrichum commune* Hedw | | I⁺ | |
| 59. | *Polytrichum strictum* Brid. | I⁺ | | |
| 60. | *Sphagnum magellanicum* s.l. *** | V⁶⁵ | V⁵ | I⁺ |
| 61. | *Sphagnum angustifolium* (Warnst.) CEO Jensen | IV³ | I⁺ | |
| 62. | *Sphagnum annulatum* Warnstorf, Carl (Friedrich E.) | | | I⁺ |
| 63. | *Sphagnum aongstroemii* C.Hartman | | I⁺ | |

**Table 1.** *Cont.*

| | Species | String, Height * of 25–50 cm | Low Hillock, Lawn, Height of 5–25 cm | Flark |
|---|---|---|---|---|
| 64. | *Sphagnum capilifolium* (Ehrh.) Hedw. | $I^+$ | | |
| 65. | *Sphagnum fallax* (Klinggr.) Klinggr. | $I^+$ | $III^+$ | |
| 66. | *Sphagnum fuscum* (Schimp.) H.Klinggr. | $III^3$ | | |
| 67. | *Sphagnum imbricatum* Hornsch. ex Russow | $II^+$ | $II^+$ | $I^+$ |
| 68. | *Sphagnum jensenii* H. Lindb. | | | $II^3$ |
| 69. | *Sphagnum lenense* H. Lindb | $I^+$ | | |
| 70. | *Sphagnum obtusum* Warnst. | $I^+$ | $II^+$ | $V^+$ |
| 71. | *Sphagnum orientale* L.I. Savicz | $I^+$ | $V^9$ | $IV^1$ |
| 72. | *Sphagnum papillosum* Lindb. | $II^2$ | $V^{70}$ | $V^4$ |
| 73. | *Sphagnum rubellum* Wils. | $V^3$ | $I^+$ | |
| 74. | *Sphagnum squarrosum* Crome | | | $I^+$ |
| 75. | *Sphagnum subfulvum* Sjors | $I^+$ | $III^1$ | $I^+$ |
| 76. | *Sphagnum subsecundum* Nees | | $I^+$ | $I^+$ |
| 77. | *Sphagnum warnstorfii* Russ. | $I^+$ | | $I^+$ |
| 78. | *Straminergon stramineum* (Dicks. ex Brid.) Hedenäs | | | $I^+$ |
| 79. | *Tomentypnum nitens* (Hedw.) Loeske | | $I^+$ | |
| 80. | *Warnstorfia exannulata* (Schimp.) Loeske | | | $I^+$ |
| 81. | *Warnstorfia fluitans* (Hedw.) Loeske | | | $I^+$ |
| | **Lichens** | | | |
| 82. | *Cetraria islandica* (L.) Ach. | $I^+$ | | |
| 83. | *Cladonia arbuscula* (Wallr.) Flot. | $I^+$ | | |
| 84. | *Cladonia furcata* (Huds.) Schrad. | $I^+$ | | |
| 85. | *Cladonia rangiferina* (L.) F. H. Wigg | $I^+$ | | |

* Height above water level in flarks. ** The coverage of the species is shown by Arabic numerals (+ means less than 1%). Species frequency classes are indicated by Roman numerals, %: V—81–100; IV—61–80; III—41–60; II—21–40; I—≤20. Empty cells indicate no species is present. *** The revision of the *Sphagnum magellanicum* herbaria samples from the Russian Far East [48] showed a wide spread of closely related species described earlier from Alaska as *S. alaskense* R.E. Andrus & Janssens [49]. In particular, this species is introduced for the inland areas of the Khabarovsk region. Later, the species *S. magellanicum* itself was divided into three [50], two of which—*S. divinum* and *S. medium*—are well-known for the mires of European Russia [51], personal observations. The first of them is widespread circumboreal species. Our specimens contain pinkish mosses bearing sings both of *S. alaskense* and *S. divinum*. The spread of this group of relative species in the Far East requires critical study, and here they are introduced generally as *S. magellanicum* s.l.

### 3.2.1. Strings

In the tree layer, only *Larix cajanderi* (0.5–2 m high, rarely up to 4 m) is found sparsely, sometimes in groups, and mostly on developed strings. The canopy rarely reaches 5% (Figure 4).

The average cover of the herb–dwarf shrub layer (HDSL) is 40%. Typical is *Betula ivaricate,* about 50 cm high, covering, on average, 10% of the string area. It sometimes covers up to 15%, being almost absent on lower strings. Of dwarf-shrubs, also found are *Chamaedaphne calyculata*, *Andromeda polifolia* and *Oxycoccus palustris* with *O. microcarpus*, *Salix myrtilloides* and *Ledum palustre* are also present.

The sedges have moderate cover of *Carex lasiocarpa* and *C. middendorffii*, which rarely occupy up to 10–15%. They are the dominant strings, with the dwarf-shrubs having less cover.

Of herbs, Sanguisorba parviflora, is present, and there is usually a low covering of *Smilacina trifolia*, *Saussurea amurensis*, *Calamagrostis langsdorfii*, *Glyceria spiculosa* and *Drosera rotundifolia*, and more typical for flarks, *Menyanthes trifoliata* and *Equisetum fluviatile*.

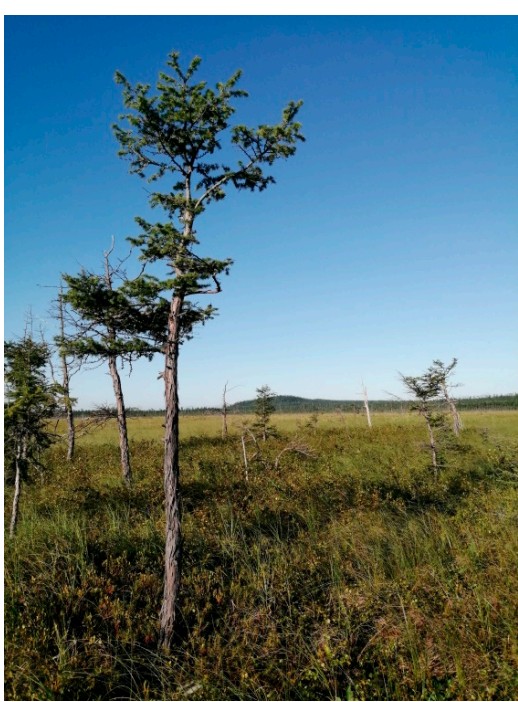

**Figure 4.** String with sparse Larix trees, A3 site.

Moss covers, on average, 75%, and consists of *Sphagnum magellanicum* s.l., with an admixture of *S. angustifolium*, *S. rubellum* and *S. fuscum*. The latter is found in small spots in only half of the sites. Sometimes *Sphagnum papillosum* is also present. Green moss (*Aulacomnium palustre*, *Dicranum undulatum*, *Pleurozium schreberi* and *Polytrichum strictum*) and lichens of *Cladonia* genus are noted in less than a half of the relevés.

### 3.2.2. Lawns

The lawns have continuous carpets of sphagnum mosses, with *Sphagnum papillosum* (65%) dominant. Common are *Sphagnum magellanicum* s.l., *S. orientale*, *S. fallax*, *S. subfulvum*, and others. In some sites, along the water edge is *Campilium stellatum*.

The HDSL is less thick than on the strings, and covers an average of 15–20%. Dominant is *Carex lasiocarpa* (10%), and frequent is *Andromeda polifolia*. Regular species with low coverage are *Betula divaricata, Salix myrtilloides*, *Oxycoccus palustris*, *Carex limosa*, *Juncus stygius*, *Rhynchospora alba*, *Scheuchzeria palustris*, *Trichophorum alpinum*, *Sanguisorba parviflora*, *Equisetum fluviatile*, *Lobelia sessilifolia*, *Iris laevigata*, *Menyanthes trifoliata* and *Parnassia palustris*, etc.

Orchids are more common on lawns than on other microforms. The most frequent are *Pogonia japonica* and *Habenaria linearifolia*. Less frequent are *Spiranthes sinensis* and *Hammarbia paludosa*.

### 3.2.3. Flarks

The flarks are covered with open water. The total compactness of the HDSL is 35%. Bladderworts (*Utricularia intermedia*, *U. macrorhiza* and *U. minor*) cover the bottom of flarks, on average by 20%, and sometimes form thickets covering up to 45%. Above-water vegetation in cases of smaller flarks is scattered over the surface and mostly along the edges in bigger and deeper ones. The most abundant are *Menyanthes trifoliata*, *Rhynchospora alba*, *Carex cespitosa* var. *minuta*, *C. lasiocarpa* and *C. limosa*. More regular here, rather than on the lawns, are *Equisetum fluviatile*, *Scheuchzeria palustris* and *Drosera anglica*. Observed are *Andromeda polifolia*, *Iris laevigata*, *Lobelia sessilifolia*, *Juncus stygius* and *Eriocaulon schischkinii*.

Moss cover is immature along the flark edges, and regularly present are *Sphagnum papillosum*, *S. orientale*, *S. obtusum* and other sphagnums. Hypnum mosses (*Campilium*

*stellatum* and *Warnstorfia exannulata*) are rare. Occasionally, there are flarks with *Sphagnum jensenii* and same botanical composition.

### 3.3. Peat Deposit Stratigraphy

Peat depth on the mire sites varies, reaching 1.3 m from the surface of flarks on A1, 2 m on A2 and up to 3.5 m on A3. In a deposit, under a dense upper layer of peat 30–50 cm deep, is frequently 0.5–1 m of water with peat-suspended solids, under which the peat density again grows, with natural layers being stowed by well-decomposed herb and herb-moss peats. The peat deposit is underlain by alluvial clay loams. Lacustrine organic remnants are not present.

In the deepest part of the deposit of the SFC of the A3 site, a parallel boring of the string and flark was conducted. In the center of the ridge (15 m wide and 30 cm high), the peat was sampled along the whole depth of the deposit—3.8 m all the way down to the mineral bottom formed by fine-grained sands and sandy loams, and from the adjacent flark down to a depth of 2 m from the water surface.

The bazal 15 cm of depositis eutrophic sedge-sphagnum peat (I on Figure 5). After this, the content of sphagnums decreases, and the main thickness of the deposit is composed by sedge (II), Scheuchzeria-sedge (III) and dwarf shrub-sedge (IV) eutrophic peats. The upper 75 cm of the deposit transitions to sedge-dwarf shrub peat (V) consisting of two layers with *Sphagnum magellanicum* s.l. and S. *fuscum*. The degree of decomposition in the lower part of the deposit amounts to 60%, varying between 30–60% along the whole thickness and decreasing to 5–15% close to the surface. The upper 30 cm contain immature peat—the tirr of *Sphagnum fuscum* over the entangled stems of the dwarf shrubs. At the depth of 50–100 cm is a water layer with peat-suspended solids. At the depth of 1.35–1.8 and 2.75–3 m, there is a slight mixture of fine-gritty sand.

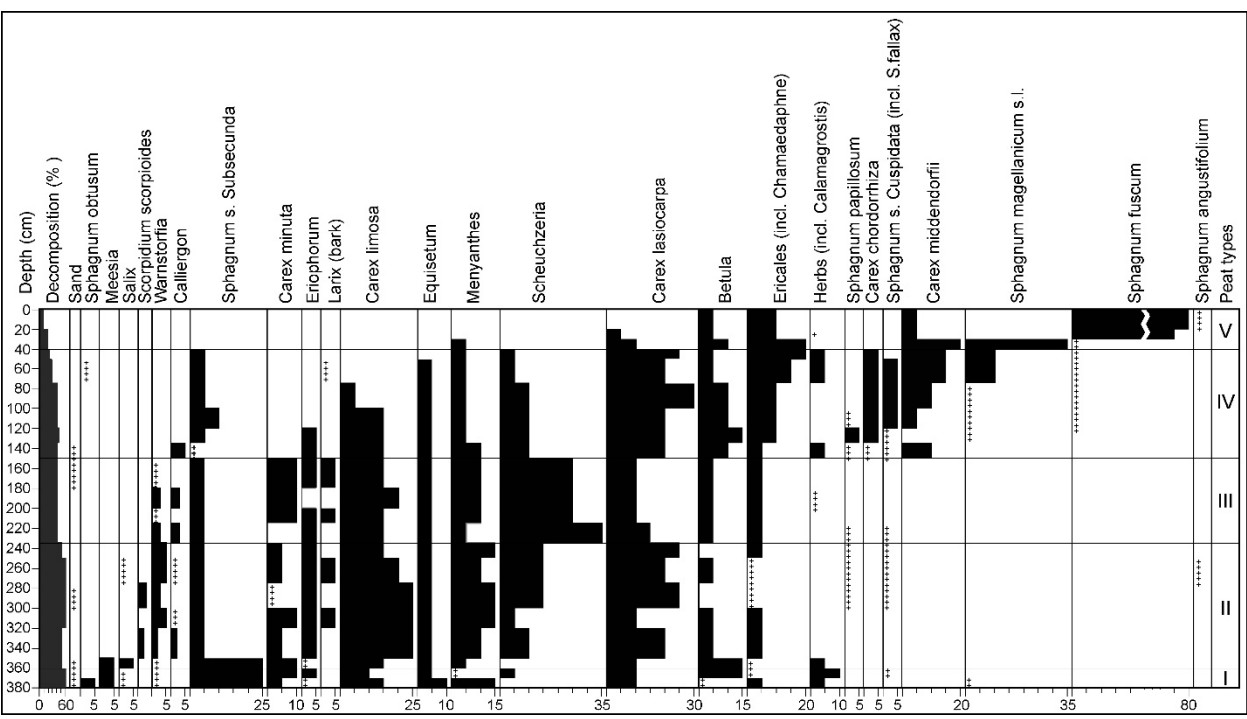

**Figure 5.** Botanical composition of peat under string. Peat types: I—eutrophic sedge-sphagnum, II—eutrophic sedge, III—eutrophic Scheuchzeria-sedge, IV—eutrophic dwarf shrub-sedge and V—mesotrophic sedge-dwarf shrub-sphagnum.

In the flark, at the peat sampling point (Figure 6), the water layer is 15 cm thick, followed below by a relatively dense 50 cm stratum of sedge peat (III). Underneath this is a 50 cm interlayer of water with peat-suspended solids. In it, just as in the underlying

peat stratum at a depth of 1.65 m from the surface, are remnants of sedges and *Sphagnum* sect. *Subsecunda* (II). Deeper, there is a stratum of *Scheuchzeria*-sedge peat (I). The degree of decomposition increases from 25% to 35–40% at the depth of 2 m. In the lower samples, a mixture of fine-grained sand is found.

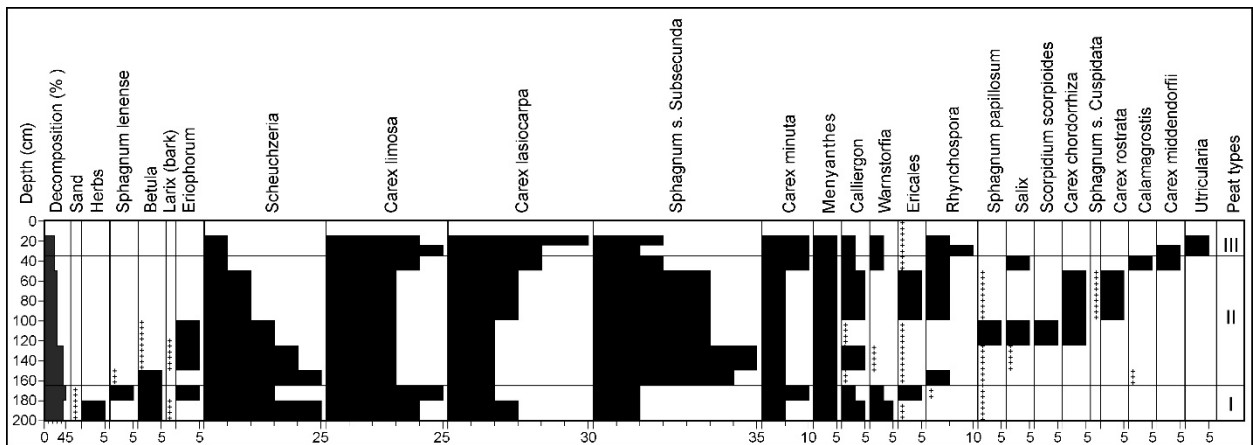

**Figure 6.** Botanical composition of peat under flark (only the upper 2 m sampled). Peat types: I—eutrophic *Scheuchzeria*-sedge, II—eutrophic sedge-sphagnum and III—eutrophic sedge.

## 4. Discussion

The mires studied correspond to the term of aapa mires in topology, surface pattern, botanical composition and stratigraphy of deposit [7]. For the territory of the Amur River region, mires had not been studied previously.

For the region, the most typical are ridge-hollow (-pool) raised bogs, which combine complex oligotrophic fractions of *Sphagnum magellanicum* s.l. and *S. papillosum* in the central areas. Minerotrophic species are absent. Closer in terms of vegetation are the quite unique complex string-flark (shallow gully) fens, in which the strings and flarks are directed along the runoff stream [38–40]. In particular, similar complexes to these consist of the stretched twisty strings and path-shaped depressions between them. They are marked as some of the most spread on the right bank area of the Amur in the central parts of mari mires [38]. Their strings are dwarf shrub-herb with *Carex lasiocarpa*, *Sphagnum magellanicum* and *S. imbricatum,* less frequent are *S. angustifolium*, *S. fuscum,* and the flarks are with *Menyanthes trifoliata*, *Carex limosa*, *Scheuchzeria palustris*, *Iris laevigata* and *Sphagnum obtusum*. V. V. Chakov [40] describes the Insky massif strings with *Chamaedaphne calyculata*, *Ledum palustre*, *Sphagnum magellanicum*, *S. angustifolium*, *S. fuscum,* flarks of *Carex limosa*, *Eriophorum polystachion*, *Sphagnum orientale*, *S. balticum* and *S. obtusum*.

Moreover, floristic compositions similar to the mires described in this study are mentioned by Prozorov [52]—sphagnum heterotrophic rain/groundwater-fed and rain-fed mires (facies) with more homogeneous tussock microrelief. Species present here include *Larix "amurensis"*, *Betula ovalifolia*, *Chamaedaphne calyculata*, *Ledum palustre*, *Carex minuta*, *C. middendorffii*, *Menyanthes trifoliata*, *Iris laevigata*, *Smilacina trifolia*, *Sanguisorba parviflora*, *Utricularia minor*, *U. intermedia*, *Glyceria spiculosa*, *Scheuchzeria palustris,* and others. In the moss layer are several species of sphagnums, among which *Sphagnum magellanicum* dominates. Hypnum mosses of flarks are absent. An important difference in the SFC characterized here is the absence of *Rhynchospora alba*, *Juncus stygius*, *Carex lasiocarpa* and *Sphagnum papillosum*.

Thus, the differences in the surface structure of the Amur region mires described here and those known from the relevant literature include differences in flora. The described mires are significantly richer in terms of species.

On the other hand, the vegetation composition of the studied mires is similar to the aapa mires vegetation of the European North, west Siberia and the Pacific coast. There

are similarities in general mire flora and the plant communities within different elements of microrelief.

Dwarf shrub–herb–sphagnum communities of strings with *S. magellanicum* s.l. are widely spread on the mires of the European North (EN) [7,9,53,54], which are ranked second in terms of prevalence on the northeast European aapa mires [7,9]. Thus, the author cites ass. *Betula nana—Carex lasiocarpa—Sphagnum magellanicum*, presenting the first stage of community change from *Sphagnum papillosum* resulting in oligotrophization and draining (in certain cases here these mosses co-dominate). Another type of string formed by *Sphagnum fuscum* [2,9,53–55] found throughout the north-west of Fennoscandia is not present on the mires considered in this article. Nor are typical species such as *Calluna vulgaris*, *Molinia caerulea*, *Frangula alnus* and *Juniperus* spp.

For the south-east of west Siberia (WS), E.D. Lapshina [56] identifies the presence of *Carex lasiocarpa* and *Chamaedaphne calyculata*, with an insignificant presence of *Sphagnum fuscum*, and no presence of *Betula nana* (vicarious with *Betula divaricata* in the Amur region) in ass. Menyantho-Sphagnetum magellanici Boč 1993—characteristic of mesotrophic and oligomesotrophic sphagnum bogs. V.A. Smagin [54] mentions similar communities with *Betula nana* for the north of the taiga zone of west Siberia. On the other hand, the combination of *Betula nana*, *Carex lasiocarpa* and *Sphagnum magellanicum* is characteristic of ass. Betulo nanae-Caricetum lasiocarpae in the composition of SFC of transition mires and mesotrophic sedge fens [56]. This association is similar to communities described here in presence of *Menyanthes trifoliata* and *Chamaedaphne calyculata*, while *Sphagnum centrale* and *S. obtusum* distinguish it. For the strings of the aapa mires of Kamchatka, *S. magellanicum* s.l. communities are not identified, although it is a regular species for blanket bog [25,57]. On the aapa strings there, *S. papillosum* dominates, and also *Sphagnum fuscum* and *S. rubellum* are present, which are not so significant for the Evoron-Chukchagir depression. To the south of Kamchatka, which is noted for regular volcanic ashfall conditions, these species are absent, and are replaced by eutrophic *Sphagnum warnstorffii*, *S. teres* and *S. squarrosum*. Taken as a whole, in the Far East there is a tendency for *Sphagnum fuscum* to be replaced by *S. magellanicum* s.l. on aapa strings when moving south and deep into the continent, similar to EN [7,9,54,58,59].

The vegetation of the strings of all specified regions brings together species of vascular plants such as *Andromeda polifolia*, *Oxycoccus palustris*, *O. microcarpus*, *Drosera rotundifolia* and *Menyanthes trifoliata*. *Betula divaricata* is vicarious with *B. nana* (EN, WS) and *B. exilis* (Kamchatka) [20] and *Ledum palustre* and *Salix myrtilloides* (EChD, EN) with *L. decumbens* and *S. fuscescens* (Kamchatka). Insignificant participation of *Pleurozium schreberi* and lichens of *Cladonia* are characteristic for all regions under consideration.

Just as in the European North and West Siberia, aapa mires frequently contain *Carex lasiocarpa* and *Chamaedaphne calyculata*. Yet these are usually absent on the mires of Kamchatka. The first one is not present on aapa in the east, and is not abundant in the south of the peninsula. The second one is noted for blanket bogs and paludified larch forests, but not for aapa.

*Carex middendorffii* and *Sanguisorba tennuifolia*, species in the east-Siberian areal, are also typical for the string–flark mires of Kamchatka. The similarity to the Kamchatka mires is also supported by the presence of larch and the absence of pine on the strings.

In the aapa mires of Kamchatka, *Myrica tomentosa* is dominant. This species is also present on the eutrophic mires of the high flood-plain of the Low Amur lowlands [52], but it is not found on the described aapa mires. The other distinction of the Kamchatka mires is the presence of some dwarf shrubs typical for the eastern tundra [24]. On the EChD, common to the SFC of Kamchatka and the EN dwarf shrubs are absent, such as *Empetrum* spp., *Vaccinium vitis-idaea*, as well as *Rubus chamaemorus*, *R. arcticus*, *Trientalis europaea*, *Trichophorum cespitosum*, and some other species.

A peculiar feature of the studied mires is the presence on the strings of *Smilacina trifolia*, *Calamagrostis langsdorfii* and *Glyceria spiculosa*.

*Sphagnum papillosum* is a characteristic species of the aapa herb-sphagnum lawns both in the west and eastwards. Thus, the *Sphagneta papillosi* formation is considered by T. K. Yurkovskaya [7,9] as the most widespread among sphagnous on the north-east European aapa. In particular, author cited ass. *Carex lasiocarpa—Sphagnum papillosum*. This species dominates the poorest aapa mires [2,9,54,55,59]. As with the EN mires, for the EChD string-flark mires, besides *Carex lasiocarpa,* typical are *Equisetum fluviatile*, *Oxycoccus palustris*, *Andromeda polifolia*, *Menyanthes trifoliata*, *Trichophorum alpinum*, *Parnassia palustris*, *Carex chordoriza*, *Carex limosa*, *Rhynchospora alba*, *Scheuchzeria palustris* and *Juncus stygius*. In the vegetation cover of the studied mires, *Molinia caerulea, Bistorta major* and*Carex rostrata* are absent; however, they are widespread in the European North.

For the south-east of West Siberia, E. D. Lapshina [56] identifies presence of species such as *Rhynchospora alba* and *Sphagnum papillosum*, and the associations with them in raised bogs. Moreover, sphagnum is exclusively inherent in oligotrophic communities with *Scheuchzeria palustris*, *Carex limosa* and *Drosera anglica*. However, the absence of *Carex lasiocarpa* alongside the minerotrophic group of species does not allow to correlate these communities with those being described in this article. Herewith, *Carex lasiocarpa* is quite widespread here, including in flarks of aapa mires, but its association with *Sphagnum papillosum*, as found in the EChD, is not present.

In contrast, for Kamchatka, present are aapa mires with lawns and strings of *Sphagnum papillosum*, but not *C. lasiocarpa* [24]. For the lawns, just as typical are the species of *Trichophorum alpinum*, *Oxycoccus palustris*, *Parnassia palustris*, *Carex limosa*, *Carex middendorffii*, *Andromeda polifolia*, *Drosera anglica*, *D. rotundifolia* and vicarious *Iris setosa*. *Rhynchospora alba*, *Scheuchzeria palustris* and *Juncus stygius* are absent.

The differences between the lawns of the EChD and the Siberian and European lawns, for the strings, are the eastern species—*Sanguisorba parviflora*, *Glyceria spiculosa*, *Smilacina trifolia* and *Carex middendorffii*, and for the flarks—*Lobelia sessilifolia* and *Iris laevigata*.

The high diversity of orchids on the aapa lawns is also common for the European North. However, only *Hammarbia paludosa* is the common species while other orchids (*Pogonia japonica*, *Habenaria linearifolia* and *Spiranthes sinensis*) have eastern ranges.

Herb flarks are a characteristic feature of aapa mires [2,7,9,32,55–57,60–62]. Common species of vascular plants of flarks throughout the Eurasian areal of aapa mires are *Menyanthes trifoliata*, *Carex limosa*, *C. chordoriza*, *Andromeda polifolia*, *Drosera anglica*, *Utricularia intermedia*, *Trichophorum alpinum* (on the EChD mires these are associated with lawns, not flarks), *Carex lasiocarpa* and *Equisetum fluviatile* (the latter two species are rare on the Kamchatka aapa). The occurrence of *Pedicularis* species is also common for the aapa flarks (*P. resupinata* on the EChD and south of Kamchatka, *P. verticillata* and *P. sudetica* in the east of Kamchatka and *P. palustris* in the EN and WS).

Flark species such as *Rhynchospora alba*, *Scheuchzeria palustris*, *Juncus stygius* and *Utricularia minor* (regular in the European North and West Siberia) are present in the EChD, but are absent on the Kamchatka aapa. *Carex laxa* in the European North is quite rare and inherent in aapa mires [63,64], was encountered only several times on the EChD, and not present in West Siberia and Kamchatka.

The EChD is similar to the Kamchatka aapa mires in its presence of *Utricularia macrorhiza*, *Lobelia sessilifolia* and *Iris laevigata* (vicarious with *I. setosa*).

Absent on the EChD yet common for Kamchatka and in the European North are *Trichophorum cespitosum*, *Eriophorum polystachyon* and *Carex livida*. This reflects the less stagnant water conditions of the studied mires. *Carex rostrata* is common for the European North, West Siberia and Koryakia [20], *Carex rotundata* and *C. rariflora* are present in Kamchatka and the northern part of the aapa areal in Europe, and *Carex cryptocarpa* is typical for the south and east of Kamchatka [23,24], yet all of them are absent in the EChD.

Peculiar features are the occurrence of *Carex cespitosa* var. *minuta*, which dominates in some flarks, and the endemic *Eriocaulon schischkinii*.

The composition of mosses of the aapa mire flarks is generally very diverse, even within one region, which reflects their critical dependence on moisture and nutrition

conditions. In the studied territory, hypnum mosses are almost absent, yet are characteristic of the rich mires of the European North, and eastern and southern Kamchatka (*Scorpidium scorpioides* and others). As a rule, among the mosses, there are same sphagnums, which is also the case for lawns. Rare along the flark borders are *Campylium stellatum*. The moss cover is mostly replaced by thickets of *Utricularia* spp., which reflects the good water flows, but not high mineralization of water.

Generally, a substantial part of the aapa mires flora of the studied territory is composed of the species characteristic of the aapa mires in the Eurasian areal (Figure 7). The other large group is common with the European mires. The species present only on the EChD mires are even fewer. Some species are similar to the flora of the Kamchatka mires. Thus, the EchD mires are more similar to the mires of the European North than to those of Kamchatka. A large number of species with a strong presence in the Eurasian areal not noted on the mires of the studied territory, which is related to a small number and lack of variety of the sample plots studied.

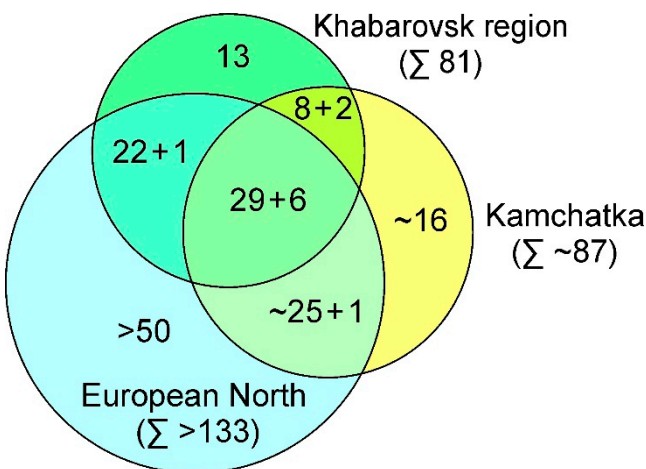

**Figure 7.** Similarity of the Khabarovsk region, the European North and Kamchatka aapa mire floras. "+" shows the number of vicarious (close) species. The data on the aapa mires flora of Kamchatka are approximate [data from 20,23,32,57]. The data on the aapa mires of the European North is approximate and based on personal observations.

The herb-dwarf shrub strings with *Sphagnum magellanicum* s.l. and the herb-sphagnum lawns of *Sphagnum papillosum* and herb flarks (which are almost without hypnum mosses and flora composition in general) are similar to the mesotrophic north-eastern European aapa [7,9,54,59]. The core species typical of these mires added up by some East-Siberian species, such as *Carex middendorffii*, *Utricularia macrorhiza*, *Lobelia sessilifolia*, *Iris laevigata* (vicarious with *I. setosa*) and *Sanguisorba parviflora* (vicarious with *S. tennuifolia*), making them similar to the Kamchatka mires. However, the similarity to the latter is not as strong, both in terms of dominant and common aapa mire species outside Kamchatka.

There is also a series of species not typical for the aapa mires outside the Amur region lowlands—*Smilacina trifolia*, *Calamagrostis langsdorfii*, *Glyceria spiculosa*, *Eriocaulon schischkinii*, *Pogonia japonica*, *Habenaria linearifolia* and *Spiranthes sinensis*. Several of them are also found on other types of mires in the Amur region [52]. Three species (*Eriocaulon schischkinii*, *Iris laevigata* and *Pogonia japonica*) are in the Red Book of the Khabarovsk Territory [65].

All the plots studied are of terrigenous genesis. The average depth of the peat deposit of the studied mires is 1.5–2.5 m. The maximum depth—3.8 m (under the string), is the largest value for the EChD mires [31].

Almost no peat dating has been conducted on the EChD. The work by Y. S. Pro-zorov [31] introduces a single reference to the work by A. V. Chernyuk [66], where there the age of a eutrophic mire with a peat deposit of 1.8 m is stated as being 3500 yr BP. However,

in similar conditions, the peatlands in the suburbs of the lake of Chlya, with a peat depth of 2.7–2.75 m have been dated at approximately 11,036 yr BP [67]. The mires of the Shantar archipelago, with a maximum depth of 3.8 m, have been dated at around 12,120 yr BP [68]. With this being said, we suppose pre-boreal peat accumulation in the origin centers of the studied mire.

The development of the deepest A3 mire started from the formation of terrigenous eutrophic herb-sedge-hypnum fen communities (Figure 3). From the very beginning, sedges have been present in the communities (*Carex limosa*, *C. lasiocarpa* and *C. minuta*), with *Menyanthes trifoliata*, *Scheuchzeria palustris* and *Sphagnum* sect. *Subsecunda* being the main peat-formers and being observed on the plot. Noteworthy is the occurrence in the lower peat stratum of the remnants of Meesia, not observed now on the studied territory. At a depth of 3.5 m is a sharp reduction in the role of *Sphagnum* sect. *Subsecunda*. Meesia disappears, although other hydrophilic hypnums (*Scorpidium scorpioides*, *Calliergon* sp. and *Warnstorfia* sp.) continue to exist in the communities. A major role is then played by herbs. Starting at a depth of 2.35 m, the sedge-*Scheuchzeria* peat starts to be deposited. The periodic occurrence of *Larix* provides evidence that the given species have existed in the communities in smaller amounts.

Starting at a depth of 1.9–2 m of the string level (which corresponds to 1.6–1.7 m from the water level in a flark) are differences in the development of string and flark communities. In the string deposit, the participation of the flark species *Scheuchzeria palustris* and *Carex limosa* declines quickly, *C. minuta* and hypnum mosses disappear, the role of *Carex lasiocarpa*, *Betula* and dwarf shrubs (*Chamaedaphne*) increases, and *Carex middendorfii* appears. Such communities—the strings forerunners—have deposited 75 cm of dwarf shrub-sedge peat.

In the flarks, at the relevant depth, *Sphagnum* sect. *Subsecunda* content rises, the role of *Scheuchzeria palustris* decreases, *Rhynchospora alba*, *Carex limosa* and *C. minuta* maintaining and the presence of *C. lasiocarpa* increases (Figure 4). In the moss cover, hypnums (*Scorpidium scorpioides*, *Calliergon* sp. and *Warnstorfia* sp.) continue to occur. Both in the communities of strings and flarks *Sphagnum papillosum*, *S.* sect. *Cuspidata* and *Carex chordoriza* appear.

In the surface peat stratum under the water are remnants of *Utricularia*. From the moment of the development of their tangles, peat accumulation in the flarks is suspended. The development of strings continues, *Sphagnum magellanicum* s.l. appears, reducing the role of sedges, and spots of *Sphagnum fuscum* start appearing.

Periodically, in the middle and lower parts, a sand mixture is noted, which is linked to the impact of seasonal floods on mire vegetation and related to water discharge from the Amgun River over the mire surface to the catchment area of the Evur river, the Evoron lake feeder. Such mineral mixtures are recorded to the depth corresponding with the commencement of SFC forming. We suppose that the development of SFC was made possible after the cessation of the direct discharge of the River Amgun waters over the mire surface. This happened as a result of the natural deepening of the Amgun riverbed and the growth of peat deposits.

Currently, regular excessive moisture on the mires do not show result in surface flooding, but in the occurrence of overwatered interlayers in the thickness of the peat under its surface stratum. Similar phenomena have already been described earlier in the region as "quagmires" in thermocarstic decline [31,38].

The peat of the modern string communities comprises only the upper 75 cm, and is mostly loose and poorly decomposed, which is related to the relative immaturity of the strings and the continuing process of changing mire patterns. The secondary nature of strings is a characteristic feature of aapa mires [9,69]. The small thickness of peat formed from ridge Sphagnum species is typical for aapa mire strings [12,58,59,62,70,71] and indicates their recent formation.

The more recent inception and active development of sphagnum communities above the sedge peat also relates to changes in mire hydrology, which may be related both to changes in climate and water catchment basin transformation [32,33,72].

## 5. Conclusions

The heterogeneous composition of the modern vegetation of the string–flark mires of the Evoron-Chukchagir depression is closely related to the vegetation of the aapa mires of the European North, West Siberia and the Pacific coast. In terms of flora composition, they are the nearest to the north-eastern European type of aapa mires. The core species characteristic of these mires all over the Eurasian areal are added up by the series of East-Siberian species.

In the absence of a marked climate-driven zone of aapa mires in the region, we relate the local development of string–flark complexes to the dynamic processes of territorial hydrology changes. The occurrence of dense and deep peat deposits of moderate and well-decomposed herb peat, as well as composition peculiarities, indicate a sufficiently prolonged period of mire formation within the depression, against a backdrop of periodic changes in the territorial character and degree of moisture. In particular, the formation of heterotrophic string-flark complexes in the place of former eutrophic fens was preceded by the cessation of flood waters from the Amgun River over the mire surface to the water catchment of the Evur River. The continued decline of erosion in the valley complex of the Amgun River and its tributaries resulted in the development of dwarf shrub-sphagnum communities on the positive forms of the microrelief.

The obvious immaturity of the current strings provides evidence of the continued active processes of mire pattern changes. It may be assumed that the area occupied by sphagnum strings and lawns within the fen will continue to grow.

**Author Contributions:** Conceptualization and methodology, V.C. and S.K.; investigation, V.C., S.K. and V.K.; writing—original draft preparation, S.K.; writing—review and editing, V.C., S.K. and V.K. All authors have read and agreed to the published version of the manuscript.

**Funding:** This research was performed within a research project of the Center for International Forestry Research (CIFOR), submitted to the Ministry of Agriculture, Forestry and Fisheries, Japan (MAFF) (LoA dated 9 June 2021; Project code CCE6510000-JPN174-DN2) and state assignments of FASO Russia (IB KarRC RAS theme No. AAAA-A19-119062590056-0; IWEP FEB RAS theme No.121021500060-4).

**Institutional Review Board Statement:** Not applicable.

**Informed Consent Statement:** Not applicable.

**Acknowledgments:** The authors express their gratitude to A.I. Maksimov (Institute of Biology of Karelian Research Centre RAS) for identifying the species of bryophytes; V.N. Tarasova (Petrozavodsk State University) for lichen identification; M.V. Kryukova (Institute of Water and Ecology Problems of Khabarovsk Federal Research Center FEB RAS) for vascular plant identification; and N.V. Stoykina (Institute of Biology of Karelian Research Centre RAS) for the botanical macrofossil analysis of peat samples.

**Conflicts of Interest:** The authors declare no conflict of interest. The funders had no role in the design of the study; in the collection, analyses, or interpretation of data; in the writing of the manuscript, or in the decision to publish the results.

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
