# Peer review of "Topology, Vegetation and Stratigraphy of Far Eastern Aapa Mires (Khabarovsk Region, Russia)"

_land, doi:10.3390/land11010096_

Round 1

Reviewer 1 Report

This paper refers to explore topology, vegetation and stratigraphy of Far Eastern aapa mires (Khabarovsk region, Russia). This study has some significance in study on peatland of Far Eastern. But there are some issues should be addressed. Thus, I suggest this paper should be revised before publication

  1. The English should be polished. The authors should asked a native English speaker to polish this manuscript.
  2. I strongly suggest authors added more sites to finish the investigation of vegetation in the study site. The current sites are very less. Thus, I suggest author added 5 sites at least.
  3. The vegetation description is very less.
  4. The authors should add paleoecology research in the sites. For example you should added the peat ages to the study profile, and the plant macrofossil should be studied.
  5. The study sites are impacted by different water vapor by the photos. I suggest the author should exclude the influence of water vapor. Or you can select the same water source type peatland and compare the vegetation features.

Author Response

1. English editing of the paper was done by a native speaker, a paleogeographer, and a CIFOR officer.

2-3. The number of descriptions provided by the authors is a representative sample, data on three spaced mire sites were used, in total 9-12 descriptions for each element of microrelief. The species composition of the communities is sufficiently homogeneous, which is confirmed by medium and high constancy classes (III-V) of almost half of the species, including the dominant communities (Table 1). Nevertheless, the geobotanical studies will certainly be continued in the future.

4. Information on paleoecology is presented in the section 3.3. Peat deposit stratigraphy characterizing the peat deposit under both positive (string) and negative (flark) relief forms.

Authors have been analyzed the macrofossil composition of the peat deposit in the deepest part, the genetic center of the bog, under the most developed ridge-hummock complexes. That is, it gives the best understanding of the development of the mire and the ridge-hummock complexes (See Figures 5, 6).

The authors do not have peat dating at the moment. It is planned to date the peat in the future. First of all, the article is focused on features of the modern vegetation of fareastern aapa mires.

5. Unfortunately, authors are not quite clear from the reviewer’s message where the water vapor is mentioned in the text. If the reviewer means the site nutrition type, then the vegetation descriptions given belong to complexes with the same nutrition type. Fig. 3a that is an oligotrophic mire margin may be misleading. Authors gave it for the general understanding of the aapa complexes and did not describe the mire of the oligotrophic type in the article. The other photos show the described ridge-hummock complexes, including ridges (Figs. 3C and 4).

Some corrections were made to the text and captions of the figures.

Reviewer 2 Report

The authors present a very interesting study of the vegetation and peat stratigraphy of String Flark complexes from the Amur region. In fact, they present the first study of this kind for this region. Therefore, the paper is very valuable to assess the botanical biodiversity for this mire type. The paper in general is well written and structured. I recommend restructuring the introduction, and shorten or replaced the geological description and better explain / introduce the research question of this study. A language and grammar check is suggested for the introduction.

Concerning the stratigraphy, question is, weather strings and flarks developed at the same position with time or did hey change position. Maybe this question can also answered and discussed in the paper.

Minor comments in the attached file.

Author Response

About the text editing: We would like to thank you for the comments. We are going to improve the paper according the comments.

The Far Eastern region is very specific in terms of geology and Holocene history and it differs significantly from the region of the most common distribution of the aapa complexes described. Therefore, we believe that an extended description of some paleogeographic features is necessary.

A conclusion about the secondary genesis of the ridges follows from the text of the article, since their surface part has an evolutionary sequence of layers beginning from the eutrophic fen. There is no indication of the prior strings’ existence on the places of modern flarks and isolated flarks on the site of modern stringes. Modern microforms evolved directly on top of the flat fen.

About the text editing: We would like to thank you for the comments. We are going to improve the paper according the comments.

The region is very specific in terms of geology and Holocene history and it differs significantly from the region of the most common distribution of the aapa complexes described. Therefore, we believe that an extended description of some paleogeographic features is necessary.

A conclusion about the secondary genesis of the ridges follows from the text of the article, since their surface part has an evolutionary sequence of layers beginning from the eutrophic fen. There is no indication of the prior strings’ existence on the places of modern flarks and isolated flarks on the site of modern stringes. Modern microforms evolved directly on top of the flat fen.

Round 2

Reviewer 1 Report

The author has greatly improved the manuscript. I agree with the publication of this paper in Land. But the language needs to be polished again.

This manuscript is a resubmission of an earlier submission. The following is a list of the peer review reports and author responses from that submission.